# A Visual Positioning Method of UAV in a Large-Scale Outdoor Environment

**DOI:** 10.3390/s23156941

**Published:** 2023-08-04

**Authors:** Chenhao Zhao, Dewei Wu, Jing He, Chuanjin Dai

**Affiliations:** 1Graduate School, Air Force Engineering University, Xi’an 710077, China; 14291024@bjtu.edu.cn; 2School of Information and Navigation, Air Force Engineering University, Xi’an 710077, China; hejing_78@163.com (J.H.); daichuanjin@126.com (C.D.)

**Keywords:** visual positioning, brain-inspired representation, feature matching, image retrieval, pose estimation

## Abstract

Visual positioning is a basic component for UAV operation. The structure-based methods are, widely applied in most literature, based on local feature matching between a query image that needs to be localized and a reference image with a known pose and feature points. However, the existing methods still struggle with the different illumination and seasonal changes. In outdoor regions, the feature points and descriptors are similar, and the number of mismatches will increase rapidly, leading to the visual positioning becoming unreliable. Moreover, with the database growing, the image retrieval and feature matching are time-consuming. Therefore, in this paper, we propose a novel hierarchical visual positioning method, which includes map construction, landmark matching and pose calculation. First, we combine brain-inspired mechanisms and landmarks to construct a cognitive map, which can make image retrieval efficient. Second, the graph neural network is utilized to learn the inner relations of the feature points. To improve matching accuracy, the network uses the semantic confidence in matching score calculations. Besides, the system can eliminate the mismatches by analyzing all the matching results in the same landmark. Finally, we calculate the pose by using a PnP solver. Furthermore, we evaluate both the matching algorithm and the visual positioning method experimentally in the simulation datasets, where the matching algorithm performs better in some scenes. The results demonstrate that the retrieval time can be shortened by three-thirds with an average positioning error of 10.8 m.

## 1. Introduction

In recent years, the development of unmanned vehicles has led to widespread research on UAV positioning and navigation in a GNSS-denied environment. Due to the advantages of strong anti-interference ability, low cost, and high accuracy, the visual positioning methods have been widely discussed. The methods utilize optical sensors to obtain environment information and process the image to estimate the UAV pose [1], which can be classified into image-based [2,3] and structure-based [4,5,6,7]. In terms of robustness and accuracy, the structured-based methods perform better in most cases. These methods usually generate a structure map in the off-line stage, which stores the reference images and matches the 2D feature points from the reference images and 3D points in the object space. As a result, each feature point in the reference images has its correspondent 3D points. In the on-line stage, the UAV can obtain some 2D-2D matches of the query and reference images, with its pose calculated using the DLT algorithm [8].

However, in large-scale outdoor scenes, the existing visual positioning methods still struggle with external condition changes. For example, illumination changes can cause shadows, highlights, and color shifts, making it difficult to detect and match accurately. Seasonal changes can affect the performance of algorithms that rely on specific features in the environment. Therefore, how to develop the neural network to learn and adapt different illumination and seasonal conditions is a crucial task in visual positioning [9].

Moreover, in large-scale outdoor scenes, both the image retrieval and feature matching will be time-consuming, due to the database which contains millions of reference images and feature points. Additionally, most matching with images in database is unnecessary. Therefore, how to make image retrieval efficient is essential for visual positioning methods.

To tackle the above problems, we propose a novel positioning system combining the brain-inspired mechanism and computer visual treatment. First, the spatial representation is constructed using place cells and landmarks. We introduce visual experiences to record the reference image and other navigation information. Then, we use a graph neural network to match the landmark via the internal relationships among the feature points. After that, the UAV can obtain the match pairs between the query image and reference image. The final step is to calculate the UAV pose using the RANSAC-based DLT solver. The main contributions of this paper are summarized as follows:Compared to existing structure-based methods, we propose a novel method to construct a cognitive map in the off-line stage. To reduce the retrieval time, we create a brain-inspired mechanism using the place cell to represent the environment. Then, we attach each reference image and landmark to the corresponding place cell. The position information can be recorded in orderly. Therefore, the UAV can narrow the retrieval scope according to the firing place cell.We proposed a graph neural network to learn the inner relationship of the feature points and perform 2D-2D matching. To improve reliability in the challenging environment, we utilize the target detection network to generate the feature points. In addition, we introduce the joint semantic confidence for matching score calculations. The score can be high only if the feature points have the same semantics. Therefore, the network can eliminate some mismatches and the matching results can be more reliable.The proposed method is evaluated on a simulation platform. We construct the cognitive map and verify the positioning method. The experimental results demonstrate that the proposed method performs better in matching and makes the positioning more efficient and accurate.

The structure of this paper is arranged as follows: Section 2 reviews the related work. The positioning system based on the brain-inspired mechanism and vision technology is illustrated in Section 3. Section 4 describes and implements details of the positioning system. Section 5 represents the experiment details on the settings, and Section 6 analyses the results. Section 7 summarizes the findings and concludes the development of the positioning system in the later research.

## 2. Related Work

The visual positioning methods in both indoor and outdoor settings have been studied extensively. The main methods can be classified into image-based and structure-based approaches. The image-based methods reflect the camera pose implicitly using the weights of a deep-learning network. Ref. [10] used an end-to-end framework to learn the pose from the database images. Therefore, the camera pose can be calculated from the corresponding color image. Refs. [11,12,13] used the neural network learning weights to regress the absolute poses, and [14,15] calculated the relative pose based on a known dataset image. However, the image-based methods are greatly affected by the challenging scenes, in which the poses calculated are unreliable. Meanwhile, the structure-based methods are hierarchical, including map construction (off-line), image retrieval, local feature matching, and pose calculation (on-line). The visual positioning system in this paper is a structure-based method.

### 2.1. Spatial Representation and Image Retrieval

Generally, most state-of-the-art methods use Sparse Structure from Motion (Sfm) [16] point clouds and images to represent the scene. The main work on spatial representation is to match the 2D feature to the 3D object points. In large-scale scenes with millions of points, the mismatches are unavoidable. Refs. [17,18] proposed a filtering to improve the match accuracy by removing the invisible 3D points. Recently, a dense 3D model scene map was used as a replacement method for spatial representation. In these methods, the 3D point can be obtained easily from the depth map, and the 2D-3D matches can be more accurate. Nonetheless, in a large-scale scene, storing the dense map is more heavyweight for memory usage than the sparse point clouds and database images. Besides, the 3D points obtained from the depth image is greatly affected by the rendering methodologies.

The existing research of visual positioning mainly focuses on feature matching and pose estimation, where few research studies pay attention to image retrieval, and even less on off-line map construction. Ref. [19] presented a coarse image retrieval, but the best-matched reference image cannot be found in this indexing scheme. Precise maps and a well-organized database contribute to improving the efficiency and accuracy of visual positioning. In other words, if the initial value of the position can be obtained, the time of the image retrieval will be independent of the scale. Therefore, aiming at the demand of image retrieval, a cognitive map combining a brain-inspired mechanism and the database is investigated in this paper.

### 2.2. Feature Matching Algorithm

According to the order of the descriptor and the detector, local feature matching can be divided into detect-then-describe [20,21], detect-and-describe [22], and describe-to-detect [23,24]. Detect-then-describe feature matching is always performed by: (i) detecting feature points, (ii) computing visual descriptors, (iii) matching these with a Nearest Neighbor (NN) search, (iv) filtering mismatches, and (v) calculating a geometric transformation. The classical pipeline developed in the 2000s is often based on SIFT, filter match pairs with Lowe’s ratio test, and calculates a transformation and rotation with a robust solver like RANSAC.

Recent feature matching, based on deep learning, is the focus of learning sparse detectors and local descriptors from the data using Convolutional Neural Networks (CNNs). To improve the network extensiveness, some research uses log-polar patches or region features to widen the vision context. Others learn to optimize matches by filtering out the outliers. These methods do not change the approach, which is still generated by the traditional algorithm, discarding the visual information and ignoring the assignment structure. Another deep learning method is based on the Transformer. Ref. [25] used self and cross attention layers to obtain both the spatial relationships of the feature points and their visual information. Ref. [26] presented a novel method that established pixel-wise dense matches at a coarse level and refined the good matches at a fine level. The graph neural network performs well in the scenes with external condition changes. However, in our application, the feature points and descriptors are similar in a region, which causes the mismatches to increase rapidly. To improve the matching performance, we add the semantics into both the feature extraction and matching score calculation in the SuperGlue [27]. The experiment demonstrates that the improved method is more accurate.

## 3. System Illustration

Figure 1 shows the visual positioning system that we propose. Different from the general positioning methods, the system needs a pre-treatment from the database so that each image will be attached to a place cell. According to the place cell firing rates, the initial position value can limit the retrieving scope. Therefore, image retrieval and feature matching will be more efficient. The whole positioning process can be divided into three steps. The first step is to imitate the brain mechanism to construct the spatial map and record the landmarks in the reference image. This way, the UAV can obtain the most relevant images attached the same place cell. Then, we use the graph neural network to establish the 2D-2D matches between the query and reference images. Meanwhile, we can select the most similar images by counting the number of matches. Finally, we introduce RANSAC to filter out the mismatches, and the UAV’s pose is calculated using the DLT solver.

As illustrated above, in the positioning system, we represent the spatial based on the place cells and landmarks exacted from the reference images. Cognitive maps can help the UAV understand the environment independently and make image retrieval more efficient. We use the graph neural network (GNN) to match the landmark. In order to make the matching more accurate, we introduce object detection in the feature points generation and introduce the semantic confidence into the matching scores calculation. Therefore, the positioning method can be improved in terms of accuracy and efficiency.

## 4. Approach

According to the system introduction, we describe the entire system from three parts: the brain-inspired spatial representation, landmark matching, and UAV’s pose estimation. The implementation details are introduced as follows.

### 4.1. Brain-Inspired Spatial Representation

In order to select the most relevant images, UCVA needs to perform a global retrieval to determine the similarity between the query and reference images in the database. However, global retrieval is complex and time-consuming. The system needs to complete many unnecessary matches with the reference images. To tackle this problem, we propose a brain-like spatial representation based on the place cells and vision sensors. Each reference image can be attached to the corresponding place cell, so that the UAV can obtain an initial position and limit the matching candidates.

An overview of the brain-inspired spatial representation method is illustrated in Figure 2. First, we establish the place cells model using the radial basis function (RBF) network, so that there is only one place cell firing at each location. Second, the vision system selects a reference image to generate landmarks using the feature points and visual descriptors. The landmark’s coordinate is calculated using the optical principle. Finally, the visual experience is established via a reference image and landmark vector, which combines the place cell and landmark to represent the environment commonly. Based on the representation, when UAV arrives at the optional location, our system will activate the corresponding place cell and narrow down the retrieval scope.

#### 4.1.1. Spatial Representation Based on Place Cells

The model of the brain-inspired representation is based on the RBF network. There are three steps to establish the spatial representation, as follows:Calculating the firing rate of the grid cell.

We adopt an oscillation inference model to mimic the grid cell, and its firing rate is calculated as follows:(1)$$R_{GC}^{i}(\vec{r})=\frac{2}{3}\left\{\frac{1}{3}\sum_{d=1}^{3}\cos\left[\frac{4\pi }{\sqrt{3}A_{i}}{\vec{k}}_{id}(\vec{r}-{\vec{\varphi }}_{i})\right]+\frac{1}{2}\right\},\;i=1,2\ldots N_{AC}$$
where $\vec{r}={\left(x,y\right)}^{T}$ is UAV’s coordinate, and $R_{GC}^{i}(\vec{r})$, $N_{AC}$ is the firing rate and number of grid cells, respectively.
Establishing the function based on the RBF network.

Via step 1, at $\vec{r}={\left(x,y\right)}^{T}$, the grid cell’s firing rate can be calculated as:(2)$$R_{GC}^{i}(\vec{r})=[r_{1},r_{2}\ldots r_{i}],i=1,2\ldots N_{AC}$$

The RBF network is trained by the input $R_{GC}^{i}(\vec{r})$ and the theoretical output place cell firing rate. Therefore, the spatial representation is constructed, which generates several place cells.
Spatial representation based on the trained neural network.

Each place cell has a firing threshold so that we can judge whether the cell is activated or not. When the UAV is at a point, the firing rate can be calculated using the RBF network. During map construction, if the firing rates are smaller than the threshold, we will repeat step 2 to generate a new place cell.

Based on the spatial representation, each position can be associated with a place cell. The approximate global coordinates of UAV can be obtained via the activated place cell.

#### 4.1.2. Landmark Generation

Landmarks show the position feature of the reference image, which will not be influenced by external condition changes. Indeed, they are naturally robust to the translation and expansion of the image, view, light changes, and clutter. To meet these requirements, each landmark consists of some feature points with visual descriptors in the reference image.

Formulation: Consider the image with a set feature point position p and associated visual descriptors d, where the positions contain the image coordinates x and y as well as a semantic confidence c, defined as $p_{i}:={\left(x,y,c\right)}_{i}$. The associated descriptors are fixed length vectors which describe the local features around the feature point. The feature points are extracted as follows.

Different from the traditional feature points extracted, our method generates feature points via a target detection network. We select the 20 common objects, such as lakes, forests, buildings, parks, etc. These objects are large enough and easy to detect at high altitudes. Once the objects are detected in the query image, the system can mark the results and mark the center point as the feature point.

As shown in Figure 3, the image is preprocessed via a target detection network. So, the targets can be marked by the anchor. We set the center of the anchor as a feature point and generated the descriptor using SURF. After that, the system can generate the feature points and fixed length descriptors on a full-sized image to represent the local feature. To make the landmark unique, we combine several feature points in the image to generate a landmark. The landmark can indicate the classes and relative position of the targets in each reference image.

#### 4.1.3. Visual Experience Generation

Visual experience consists of the reference image and landmark vector. We embed the place cell and landmark into a high dimensional vector, which is defined as:(3)$$landmark\;vector=\{{\mathrm{Fr}}_{id},{\mathrm{Pc}}_{id},p_{ucav},N_{p},(p_{id},p_{pix}^{0},p_{real}^{0}),\ldots ,(p_{id},p_{pix}^{N_{P}},p_{real}^{N_{P}})\}$$
where ${\mathrm{Fr}}_{id}$ and ${\mathrm{Pc}}_{id}$ is the ID of reference image and place cell, respectively. $p_{ucav}$ is the UAV’s pose. $N_{p}$ represents the total number of feature points in the landmark. The coordinate of each feature point in both the pixel and real worlds are computed and saved as $\left(p_{id},p_{pix}^{N_{P}},p_{real}^{N_{P}}\right)$. Based on the landmark vector, each frame attaches to one place cell. Once the landmark in the query image is matched, the navigation information can be obtained from the landmark vector to estimate the UAV’s pose.

### 4.2. Landmark Matching

The second major block of the visual positioning system is landmark matching, which selects strong matches both in query and reference images. When asked to match a known ambiguous landmark, humans usually look back-and-forth at both images: they sift through the matching buildings, examine each, and look for spatial and visual relationships that improve the matching accuracy. To imitate human behavior, we propose a novel graph neural network where the learning relationship between the feature points is in the same landmark. The network performs well in image matching.

Through Section 4.1.2, both images have *M* and *N* feature points, indexed by A:={1,…M} and B:={1,…N}, respectively. For facilitating downstream tasks and a better localization, each correspondence should have a confidence value. We define a partial soft assignment matrix $P\in {\left[0,1\right]}^{M\times N}$ as:(4)$$P1_{N}\leq 1_{M}\;\mathrm{and}\;P1_{M}\leq 1_{N}$$

According to the Superglue, the GNN aims at predicting the assignment P from two sets of landmarks. As shown in Figure 4, the neural network consists of two major layers: an attentional graph neural network and an optimal matching layer. The first layer aims at creating more powerful representations f using self- and cross-attention layers (repeated *L* times).

The detailed steps of the GNN are as follows:Feature point encoding via a multilayer perceptron (MLP)

To enable the GNN to later reason about both the appearance and position jointly, we embed the feature point position into a high dimensional as:(5)x(0)i=di+MLP(pi)
Learning relations via alternating self- and cross-attention layers.

We construct a single complete graph whose nodes are the feature points of both *A* and *B*. So, the graph has two types of edges: self-edges and cross-edges. These edges can connect feature point *i* to all the others in both *A* and *B*. At each layer, the GNN network computes an updated representation by simultaneously aggregating messages across all the given edges for nodes, as follows:(6)x(l+1)iA=x(l)iA+MLP([x(l)iA||mε→i])
where the message $m_{\varepsilon \to i}$ is the result of the aggregation from all feature points $\left\{j:(i,j)\in \varepsilon \},\;\varepsilon \in \{\varepsilon_{\mathrm{self}},\varepsilon_{\mathrm{cross}}\right\}$. [⋅||⋅] denotes concatenation. A fixed number of layers *L* with chained alternatively aggregates along the self- and cross-edges. As such, starting with l=1, if *l* is odd $\varepsilon =\varepsilon_{\mathrm{self}}$ and if *l* is even $\varepsilon =\varepsilon_{\mathrm{cross}}$.
Final matching descriptors are linear projections:


(7)
fiA=W⋅x(L)iA+b, ∀i∈A


Via the method above, the GNN can obtain a powerful representation f, the optimal matching layer, which produces a partial assignment matrix. In the graph matching formulation, P can be obtained by computing a score matrix $S_{i,j}$ recording all the possible matches and maximizing the value of $\sum S_{i,j}P_{i,j}$ under the constraints in Equation (4). Different from the SuperGlue, we introduce the semantics into the score matrix calculations.

In the process of generating the feature points, a vector $c_{i}^{A}$ is used to represent the semantic information of them. The dimension of $c_{i}^{A}$ is set as 20, and each value represents the confidence of the corresponding objects.

As such, we express the pairwise score as the similarity of the matching descriptor:(8)$$S_{i,j}=c_{i,j}<f_{i}^{A},f_{j}^{B}>,\forall (i,j)\in A\times B$$
where <⋅,⋅> is the inner product. *A* and *B* are the picture names. *i* and *j* are the index of the feature points in an image. $c_{i}^{A},c_{j}^{B}$ is the semantic confidence vector of the feature points. $c_{i,j}$ is the mixed confidence of the match pairs. We obtain a new vector $c_{i,j}^{AB}$ by multiplying the corresponding terms of $c_{i}^{A}$, $c_{j}^{B}$ and selecting the maxim $c_{i,j}$ in the vector as the matching semantic. Therefore, only when the points have the same semantics, the pairwise score can be high. This operation can make the feature points with the same semantic label have a strong relationship. The network can utilize the semantic to eliminate some mismatches. After that, the optimal layer uses the Sinkhorn algorithm to find the optimal partial assignment P. The implement details are described in [27].

### 4.3. Position Calculation

Based on the first two steps, UAV can match the query image *U* with the most relevant image *R* using the method proposed above. As shown in Figure 5a, we assume a feature point $\left(x_{r},y_{r}\right)$ in the reference image is matched to the feature point $\left(x_{q},y_{q}\right)$ in the query image. According to the landmark vector, this match corresponds to a 3D point P $\left(x_{p},y_{p},z_{p}\right)$ in the object space. Hence, we calculate the 6-DOF pose via the DLT algorithm.

As shown in Figure 5b, a pair of matching landmarks can generate several matching points. But, there are many mismatches causing the pose calculation to be inaccurate. To improve the performance of the pose estimation, we use an efficient RANSAC framework to filter the outliers of the match pairs. After filtering, the pose is calculated as follows:

For each match pair, we can generate two linear constraints:(9)$$\begin{array}{l} t_{1}^{T}P-t_{3}^{T}Px_{vi}=0\\ t_{2}^{T}P-t_{3}^{T}Py_{vi}=0 \end{array}$$ where **R** is the rotation matrix, and $t={\left(t_{1},t_{2},t_{3}\right)}^{T}$ is the translation matrix. Assume there are *N* match pairs, so that we can obtain the equations, as shown below:(10)$$\left(\begin{array}{ccc} P_{1}^{T} & 0 & -x_{v1}P_{1}^{T}\\ 0 & P_{1}^{T} & -y_{v1}P_{1}^{T}\\ \ldots & \ldots & \ldots \\ P_{N}^{T} & 0 & -x_{vN}P_{N}^{T}\\ 0 & P_{N}^{T} & -y_{vN}P_{N}^{T} \end{array}\right)T=AT=0$$

According to the solving requirements, there needs be six match pairs at least. And, when the number of match pairs is six, we can obtain the R and t directly. Otherwise, we calculate them by the least square. The details of the solving process are described in [28]. The rotation matrix R and translation matrix t can be calculated as follows:(11)$$R=\pm UV^{T},\;t=\beta V_{4}\;\beta =\pm 3/tr(\sum )$$

Therefore, we can obtain the camera pose in the global coordinate, as follows:(12)$$P_{cam}=RP_{ref}+t$$

So, the UAV pose can be obtained as follows:(13)$$P_{UCAV}=R_{1}P_{cam}+t_{1}$$
where the $R_{1}$ and $t_{1}$ are the rotation and translation between the UAV coordinate and camera coordinate.

## 5. Experiment Settings

### 5.1. Simulation Environment

In order to verify the efficiency and accuracy of the proposed positioning system, we compare the performance of the system with other visual positioning methods. The experiments were carried out in a simulation platform, which was constructed by the authors. In the platform, the UAV pose, flight altitude, airspeed, and query image can be obtained in real time.

Besides, the area of the outdoor scene is 30 km^2^, which contains the airport, country, lake, hill, etc. The platform is constructed using the real terrain and satellite photos, and the real terrain uses the mesh generated via surveying and mapping. To improve the precision of the virtual scene, we use a 3D model to reproduce the real scene at some feature abundant place such as the airport and buildings in urban area, as shown in Figure 6. Through these operations, the platform can replicate the real scene to more than 80%.

### 5.2. Experiment Details

The cognitive map construction and feature matching experiment were implemented using PyTorch on NVIDA RTX 3090 GPU. The following experiment was set as follows:

Preparatory condition: ① The whole process contains three coordinates: the global coordinate, the camera coordinate, and the UAV coordinate. Especially, we set the take-off point as the origin of the global coordinate. ② The camera is fixed to the UAV, so the rotation and translation matrices are constant. ③ In order to generate enough landmarks, the relative altitude of UAV is set to (200 m and 300 m). ④ Each feature point in the landmark has been mapped to a region in the object space. We used the coordinates of the region center. To simply the pose estimation, we attach the center’s global coordinates in the region to the correspondent feature points. ⑤ In order to confirm the positioning accuracy, we utilize the true coordinate in the object space to correct the 3D points calculated using the optical principle.

## 6. Results and Analysis

### 6.1. Cognitive Map Construction

Before the positioning experiment via the vision sensor, we need to construct the cognitive map of the simulation environment. The UAV explored the whole scene and generated 36 place cells with the correspondent feature points and reference images. After that, we calculate the global coordinate of the feature points. Ideally, the place cell firing field is circular. As shown in Figure 6, some of the 3D feature points and circles are drawn in the global coordinate. We use blue circles to represent the firing field. All feature points in each circle are attached to the corresponding place cell. Based on the cognitive map, only one place cell will be activated, while the UAV is in somewhere. As a result, we can obtain an initial position to calculate the accurate value later.

After cognitive map construction, we mainly verify the image retrieval and vision positioning method. The details and results are as follows (Figure 7):

### 6.2. Image Retrieval

To evaluate the efficiency of the image retrieval algorithm used in positioning methods, we construct different cognitive maps and databases which contain different numbers of images, and the results of the experiment could represent the performance of the retrieval algorithm. We introduce the retrieve time to evaluate the performance of the algorithm efficiency. The retrieval time is the time to search for the most related image in the database. This time depends on the matching time and retrieval method.

The number of images in the database is 2000, 4000, 6000, 8000, and 10,000, respectively. In order to reduce random errors, 20 query images were randomly selected for the experiment, and each image was retrieved 10 times. In Table 1, the running time is expressed with the average time of image retrieval.

The method (no map) uses the single-layer clustering algorithm (SLCA) and retrieves the reference image one by one. The Hierarchical Clustering-based Image Retrieval (HCIR) divides the process into three parts and retrieves the image via feature points. As can be seen, our method has a significant time priority compared with the traditional retrieval method. This is mainly because the proposed method can eliminate the most irrelevant images before the matching process, reducing the retrieval times before each matching process. When the database image is 2000, our methods (with map) only need to perform matches up to 51. In this case, the SLCA almost reaches 2000 times. Compared to HCIR, the proposed method also shortens the retrieval time by two-thirds. This reduction limits the retrieval’s time-consuming to 100 ms, which can improve the method’s real-time performance.

However, the retrieval time linearly increases, along with the growth of reference image, if the place cells remain unchanged. As shown in Table 2, there is a contradiction between map accuracy and retrieval efficiency. The higher accuracy of the map contains more reference images, resulting in a decrease in retrieval efficiency. With the increase in reference images, the environment can become more detailed, and the UAV pose can be calculated more frequently. According to the simulation results, when the total number of reference images is 6000, both the efficiency and accuracy are within a reasonable range, meeting the commands of UAV positioning.

### 6.3. Feature Matching

To evaluate the matching performance, we use both the SuperGlue and the proposed method in the simulation platform. We select some representative matching results to demonstrate the advantages of the proposed method.

In some scenes, the SuperGlue performs as well as the proposed method, as shown in Figure 8a,c. In the experiments, we obtain the same matching results via both methods. However, there are some feature points with familiar descriptions which makes the SuperGlue generate some mismatches. The proposed method can eliminate these mismatches, like the red lines in Figure 8b,d. In Figure 8b, the feature points have different semantic confidences. As shown in Table 3, the points A and B obtain high matching scores in the SuperGlue, while the proposed method only gives 0.13. This is because the matching scores areinfluenced by the semantic confidence of the feature points in the proposed method. Only when the feature points have the same semantics, the matches can obtain a high score. In Figure 8d, blue lines represent the relations of the feature points and are recorded as the adjacency matrix in the algorithm. As a result, we can also examine the results by comparing the matrix, as shown in Figure 8d.

It is essential to obtain robust and accurate image-matching results efficiently in outdoor conditions. The GNN network can utilize self- and cross-information to learn the relationships of the feature points, as shown in Figure 8a,c. But there are still mismatches in the outdoor environment. The proposed method introduces the semantics into both the feature points generation and the matching score calculation. Using the approach, the method can correct some mismatches in Figure 8b,d. The simulation results demonstrate that the proposed matching method performs well in the outdoor environment.

### 6.4. Vision Positioning

The trajectory of the UAV and single-positioning points are shown in Figure 9. The flying process consists of the cruise, climb, and glide. The speed is set as 60 m/s. The total time of simulation is 1150 s and positioning cycle is 20 s. The results of the experiment contain: ① positioning at different points; and ② positioning throughout the flying. The accuracy of positioning is defined by the orientation and position error. The orientation error $r_{error}$
is calculated as follows: $r_{error}={\left\|r_{est}-r_{gt}\right\|}_{2}$, in which $r_{est}$ is the estimation angular vector, and $r_{gt}$ is the ground truth angular. The positioning error $p_{error}$ is defined as the Euclidean distance: $p_{error}={\left\|p_{est}-p_{gt}\right\|}_{2}$, in which $p_{est}$ is the estimation position and $p_{gt}$ is the ground truth position.

We also analyzed the flying positioning throughout. We repeat flying in the same trajectory 10 times and calculate the arithmetic average. To demonstrate the performance of the proposed positioning method, two typical visual positioning methods (i.e., the SIFT + NN and the SIFT + SuperGlue) are selected and implemented. According to Table 4, the positioning errors are calculated, and the proposed method has the minimum error in most cases (average 10.8 m, maximum 12.2 m). The data with improvements is compared to Table 4 and are highlighted in black bold. The results show that our positioning methods perform best in most cases, confirming the semantic information can help the network eliminate the mismatches effectively. These results further highlight the effect of our matching methods.

To enhance the credibility of the results, we also report the percentage of the points positioning with the varying thresholds on their positioning error during the whole flying process. The results consider only the points with an error of (20, 15°) or less. As shown in Table 5, the proposed method performs more accurately in all three stages. It can limit errors to 89.2% in (20 m, 15°) and 13.4% in (10 m, 5°). The results demonstrate that the proposed method can obtain more precise poses in most scenes.

Besides, in some feature abundant points, both the SuperGlue and the proposed method perform well in the matching process. But there is still a gap in the pose estimation. The first two methods’ error does not only depend on the matching method but also on the coordinate acquisition. Due to the generation of feature points, these methods obtain an inaccurate 3D coordinate in the space. Besides, the proposed method generates feature points based on the targets with specified coordinates. So, the proposed method performs better in positioning.

## 7. Conclusions

In this paper, a novel map construction method is presented, in which landmarks and reference images are attached to the corresponding place cells. In this approach, the UAV can narrow down the retrieval scope according to the firing place cell and improve retrieval efficiency. Besides, we use the GNN to learn the inner relationships of the feature points and to introduce the semantics into the matching score calculations. Using this, the proposed method can perform better in external condition changes than the existing methods. Our experiments in the simulation environment show that our proposed system outperforms the competing methods in efficiency and accuracy. Moreover, the total time for positioning can be shortened by around three-thirds with higher precision. The proposed method can improve the efficiency and decrease the influence of illumination and seasonal changes. However, with the increase in area, the maps and weight files will occupy more space, leading to the increase in computation cost. Therefore, in future work, we will add more navigation information to improve the complexity of the cognitive map. Furthermore, we will use dynamic convolution to design a neural network with a smaller weight file, which can adapt to large outdoor environments.

## Figures and Tables

**Figure 1 sensors-23-06941-f001:**
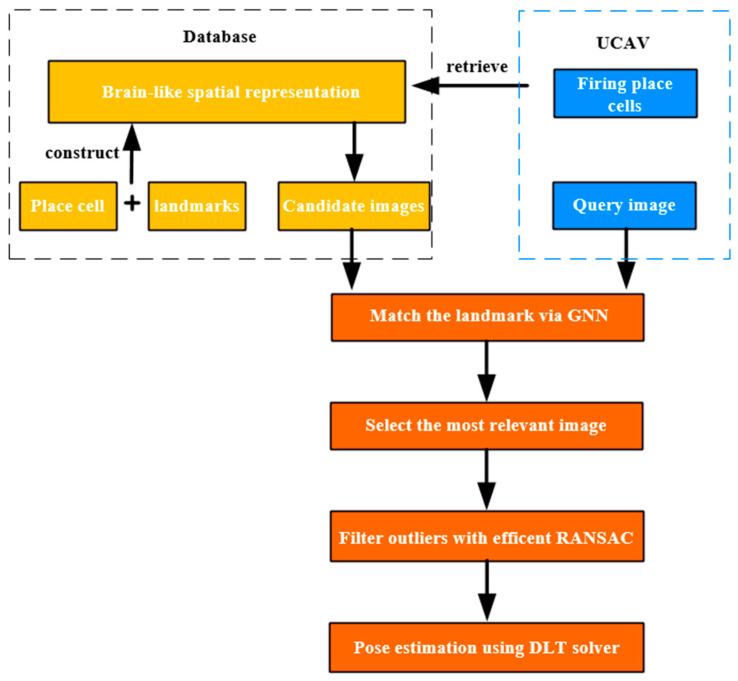
Illustration of the visual positioning system. The hierarchical positioning method pipeline is divided into three parts. Yellow squares show the spatial representation based on the place cell and landmarks in the off-line stage. Blue squares show the UAV’s real-time perception, and orange shows the feature matching process and positioning via the DLT solver in the on-line stage.

**Figure 2 sensors-23-06941-f002:**
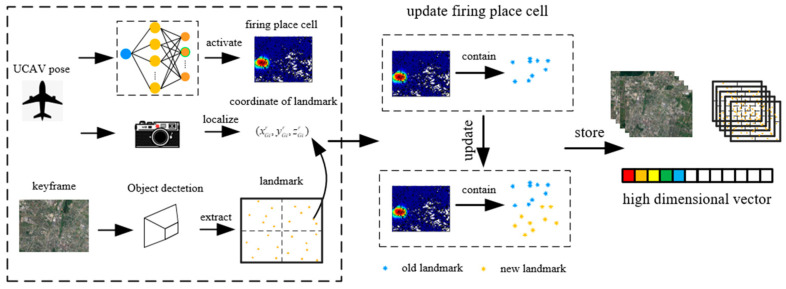
The construction of spatial representation based on the vision sensor and place cell.

**Figure 3 sensors-23-06941-f003:**
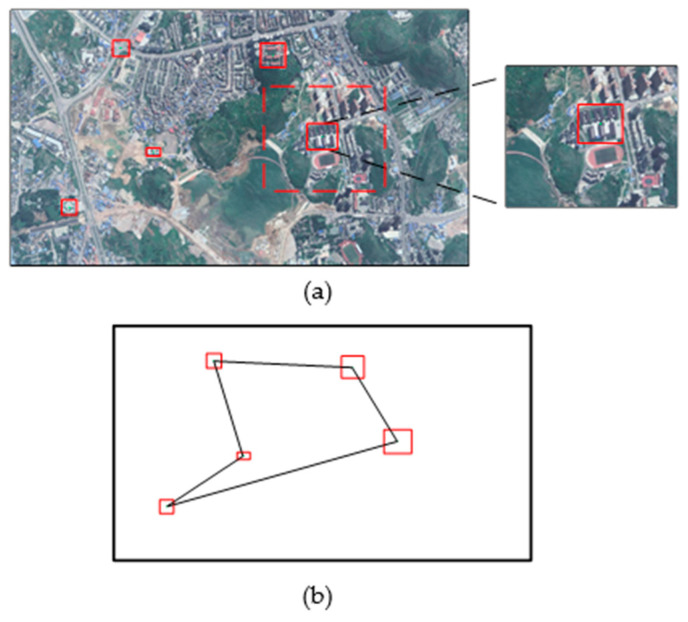
The landmark generation via target detection. (**a**) Image processing. We set the center of the anchor as the feature point and generated visual descriptors using SURF. (**b**) Landmark generation. The red boxes are the results of object detection.

**Figure 4 sensors-23-06941-f004:**
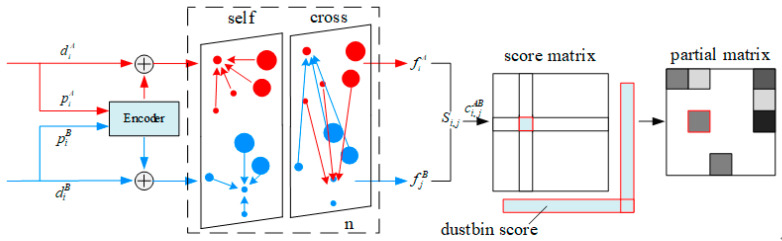
Feature matching using the graph neural network.

**Figure 5 sensors-23-06941-f005:**
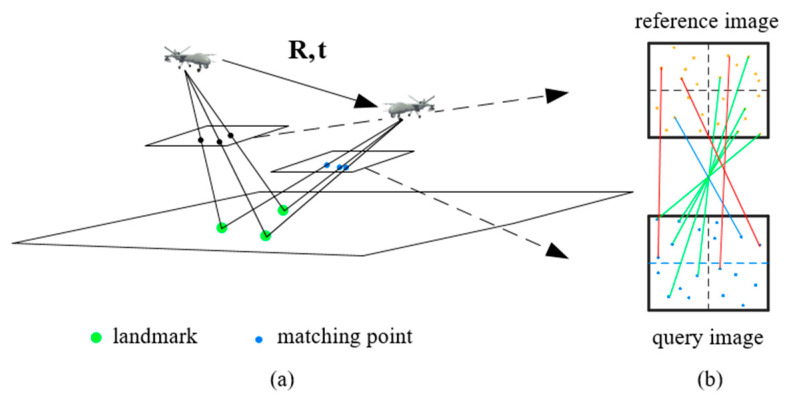
The principle of pose estimation and the image coordinate. (**a**) The UAV obtains the coordinates of the landmark and match pairs in both the reference and query images. (**b**) It shows the results of the matching algorithm. The orange and blue dots are the feature points in reference image and query images, respectively. Green lines represent the matches that are accurate and red lines are mismatches. Blue lines represent the matches score between “accurate” and “mismatches”.

**Figure 6 sensors-23-06941-f006:**
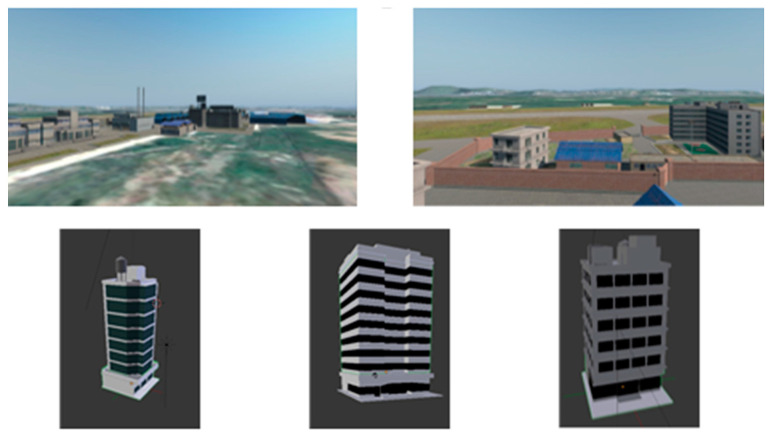
The simulation scene and some 3D models.

**Figure 7 sensors-23-06941-f007:**
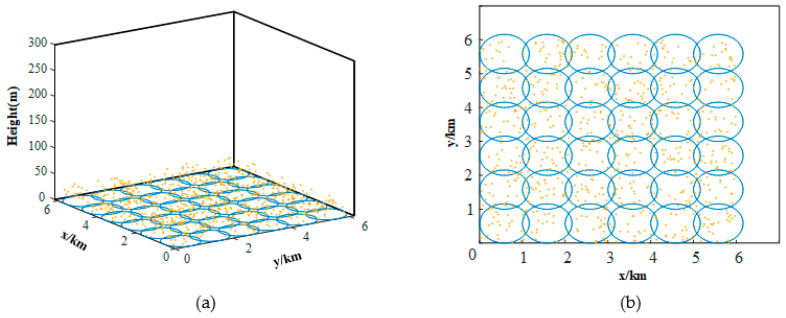
We show the results of spatial representation and some of the 3D feature points in both 3D (**a**) and 2D (**b**) perspectives. In these pictures, each blue circle means the place cell’s firing area, and yellow marks are the 3D points (known coordinates) in the object’s space.

**Figure 8 sensors-23-06941-f008:**
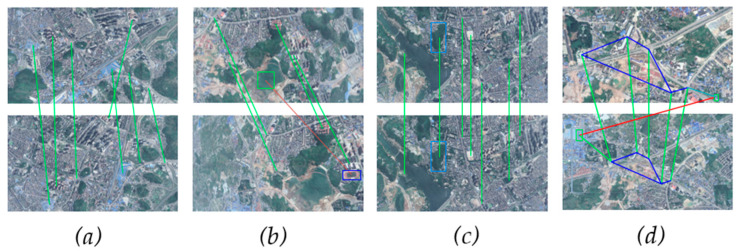
The results of the proposed matching methods. In the figure, the green line shows the matches that are accurate. The red lines represent mismatches. Different color squares represent different target types. In (**a**,**c**), both SuperGlue and proposed method perform well. In (**b**), green box and blue box mean different semantics. In (**d**), the color lines mean the relations of the feature pionts are in the same image.

**Figure 9 sensors-23-06941-f009:**
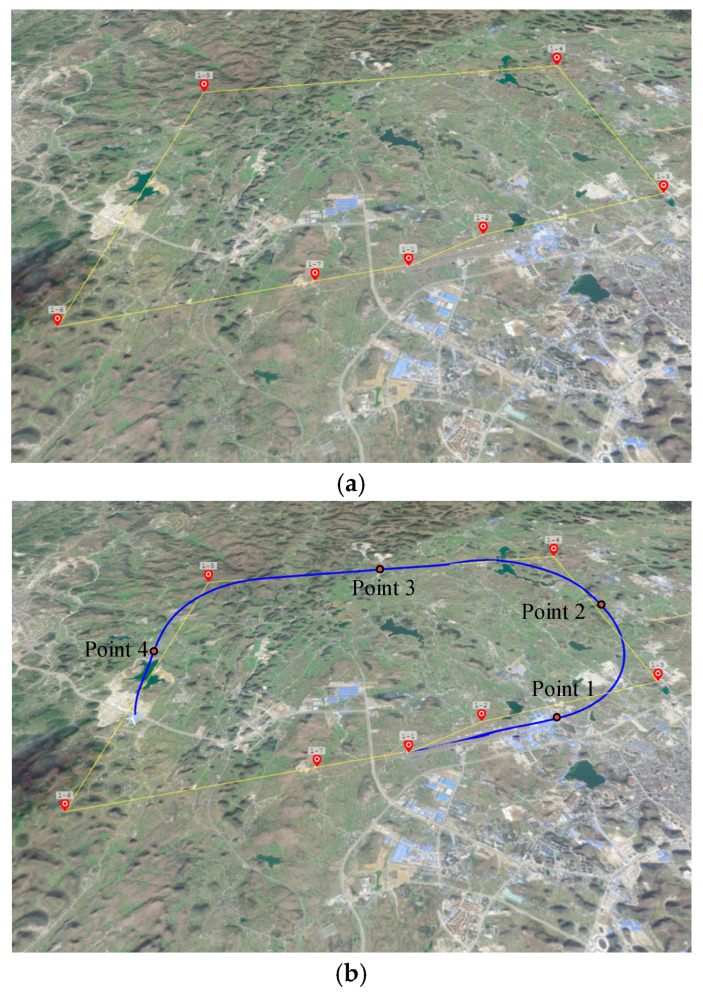
Planning and real trajectory of UAV in the simulation platform. (**a**) The planning trajectory of UAV in the simulation platform. (**b**) The real trajectory of UAV in the simulation platform.

**Table 1 sensors-23-06941-t001:** Retrieval time comparison of the proposed method using the traditional hierarchical localization baseline (unit: ms).

Total Number of Images	Key Points of Image	HCIR	The Method (No Map)	The Method (with Map)
		Running Time	Running Time	Running Time
2000	5	64.08	1521.45	42.65
	10	98.65	2106.78	59.21
	15	112.41	3463.12	90.33
4000	5	131.36	2412.44	79.54
	10	187.12	3947.12	112.86
	15	231.44	6101.47	164.41
6000	5	197.36	4245.13	119.45
	10	288.85	6208.34	175.35
	15	340.41	9169.12	262.24
8000	5	264.68	1886.64	140.74
	10	375.23	7456.12	224.24
	15	461.11	11,423.08	342.68
10,000	5	395.76	6541.78	183.37
	10	480.24	9576.92	276.22
	15	561.46	12,104.41	391.48

HCIR: hierarchical clustering-based image retrieval.

**Table 2 sensors-23-06941-t002:** The average positioning errors with different numbers of reference images.

The Number of Images	2000	4000	6000	8000	10,000
Error Dis. [m]/Orient. [deg]	27.3/12.1	20.4/10.9	13.6/10.1	10.2/9.7	9.8/9.2
Running time [ms]	59.21	112.96	175.35	224.24	276.22

**Table 3 sensors-23-06941-t003:** Some values of feature points in Figure 8b.

Point	Lake Cofiendence	Forest Cofiendence	MS (SupreGlue)	MS (Proposed)
A	0.68	0.16	0.82	0.13
B	0.23	0.71		

**Table 4 sensors-23-06941-t004:** Pose estimation error of the single point.

Check Point	Error (SIFT + NN)	Error (SIFT + SuperGlue)	Error (This Method)
	Dis. [m]/Orient. [deg]	Dis. [m]/Orient. [deg]	Dis. [m]/Orient. [deg]
1	19.8/18.9	14.5/11.5	8.9/6.9
2	18.8/15.6	17.4/14.2	10.6/8.7
3	20.7/17.8	10.9/10.4	11.4/10.2
4	25.6/19.2	15.3/10.2	12.2/11.2

**Table 5 sensors-23-06941-t005:** Pose estimation during the whole flying process.

Method	Positioning Queries (%)		
	10 m, 5°	15 m, 10°	20 m, 15°
SIFT + NN	8.1	32.4	57.4
SIFT + SuperGlue	12.1	57.6	77.6
Proposed method	13.4	62.2	89.2

## Data Availability

Not applicable.

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
