# Peer review of "A Visual Positioning Method of UAV in a Large-Scale Outdoor Environment"

_sensors, 2023, doi:10.3390/s23156941_

Round 1

Reviewer 1 Report

1. Begin the introduction by providing a brief overview of UAV visual positioning and its importance in outdoor environments. Explain the challenges faced by existing methods in handling illumination and seasonal changes. 2. Clearly state the novelty of your proposed method in comparison to existing structure-based methods. Highlight the key contributions and advantages of your approach, such as improved reliability in challenging outdoor conditions and reduced retrieval time. 3. Provide more detailed explanations of the steps involved in your hierarchical visual positioning method, including map construction, landmark matching, and pose calculation. Elaborate on how the brain-inspired mechanism and landmarks are combined to construct the cognitive map, and how the graph neural network is utilized to learn the feature points' inner relations. 4. Describe in more detail how the semantic confidence is incorporated into the matching score calculation. Clarify what semantic information is used and how it helps improve matching accuracy. 5. Provide a clearer explanation of how the system analyzes the matching results in the same landmark to eliminate mismatches. Describe the specific criteria or techniques used for this analysis. 6. Provide a rationale for choosing the P3P solver for pose calculation. Explain its advantages and why it is suitable for your proposed method. 7. Provide more information on the simulation datasets used for evaluating both the matching algorithm and the visual positioning method. Explain how these datasets represent real-world outdoor environments. Include details on the metrics used to assess the performance of the methods. 8. Interpret and discuss the results more comprehensively. Provide insights into the strengths and weaknesses of the proposed method compared to the existing approaches. Highlight any limitations or potential areas for improvement. 9. Summarize the main findings and contributions of your research more effectively. Emphasize the significance of your proposed method in addressing the limitations of existing approaches and its potential impact on UAV visual positioning in large-scale outdoor environments. 10. Check the manuscript for grammar, clarity, and coherence. Ensure that all statements and claims are supported by evidence and properly referenced where necessary.

Check the manuscript for grammar, clarity, and coherence. Ensure that all statements and claims are supported by evidence and properly referenced where necessary.

Reviewer 2 Report

The paper presents a research work on visual positioning of UAV in large-scale outdoor environment. The topic seems to be original.

The authors propose a new hierarchical visual positioning method using map construction, landmark matching and pose calculation techniques. It seems to provide appropriate up-to-date descriptions and comparisons with some of the well-known methods. The proposed method was verified through simulations.

The quality of the paper is about average. English language must be taken care. The editing and the format of the text must also be taken care in some cases. In page 9, there is a missing text (“…, which constructed by.”).

Contribution No3 is not clear.

In reference 27 the authors are not well stated.

The authors, based on simulation results, conclude that their method performs better in positioning compared to SuperGlue method. It would have been useful to be stated whether the SuperGlue has already been validated with real experimental tests of similar cases. Although an interesting question could be whether the proposed method would indeed outperform SuperGlue in real case studies, of course there is no need to be answered since the authors have worked with simulations only.

A general discussion of the results obtained, either as a separate paragraph or subsection would have been useful in section 6.Results and Analysis. Otherwise, perhaps the conclusions could be improved in order to emphasize the contributions, and be consistent with the results presented.

English language must be taken care.

Round 2

Reviewer 1 Report

Author have made all the suggested changes. Hence paper is accepted in its present form.

Reviewer 2 Report

Still, in reference 27 the authors are not well stated.

English must be taken care.